# Reconstruction and Visualization of Landslide Events Based on Pre- and Post-Disaster Remote Sensing Data

Zhaolin Luo [1], Jiali Yang [1], Bolin Huang [2,*], Wufen Chen [1], Yishan Gao [1] and Qingkui Meng [1]

1 Pearl River Water Resources Research Institute, Guangzhou 510611, China
2 Hubei Key Laboratory of Disaster Prevention and Mitigation, China Three Gorges University, Yichang 443002, China
* Correspondence: bolinhuang@aliyun.com; Tel.: +86-18607155502

**Abstract:** This paper proposes a method to reconstruct and visualize landslide events based on pre- and post-disaster remote sensing data. The proposed method establishes the dynamic equations of the landslide evolution process and calibrates the model parameters based on pre- and post-disaster remote sensing data. Based on the calibrated dynamic equations, we reconstruct and simulate the historical landslide process and visualize the landslide evolution. The experimental results show that our method could dynamically and realistically reconstruct and visualize the landslide evolution process. Moreover, the landslide process simulation can also detect the maximum depth, maximum sliding speed, maximum momentum, and other indicators during the evolution process, and the visualization results can be used for subsequent hazard assessment, engineering implementation, and other applications.

**Keywords:** landslide; dynamic equations; reconstruction; visualization; remote sensing data





## 1. Introduction

Landslides are one of the most widespread and frequent geological disasters in nature. They pose a significant threat to hydraulic engineering due to the instability of reservoir banks and the suddenness of the landslides. Landslides and the corresponding secondary disasters cause tens of thousands of deaths and incur substantial economic losses each year. The occurrence of landslides has had a significant impact on people's lives and the economy, and landslide safety has become the top priority in the field of hydraulic engineering geological disaster prevention and control. The impact of landslides near water conservancy projects is shocking, causing a large number of casualties and significant economic losses worldwide every year [1]. For example, in January 2013, a landslide occurred at the Zhaojiagou Reservoir in Zhaojiagou Village, Yunnan Province, China, killing 46 people and injuring 2 others. In addition, it buried 14 buildings, and approximately 210,000 cubic meters of debris tumbled down the steep hill. As of 4 p.m. on 12 January 2013, 43 dead individuals were discovered, and 5 individuals were still missing. Similarly, on 16 May 2021, a large-scale landslide occurred at the Daxingzhai Reservoir in Xingtian Village, Shuangtang Street, Jishou City, China, killing three people and causing a direct economic loss of 4.25 million yuan. On 9 August 2019, a landslide occurred around the Motama Reservoir in Bangzhen, Myanmar, killing 70 people and destroying all 27 houses. On 29 October 2022, landslides occurred in the reservoir area of Maguindanao del Norte, the Philippines, and the death toll in the Bangsamoro region exceeded 155 people. Landslides have become a major factor in the safety of hydraulic projects around the world. This geological disaster causes economic losses by destroying water conservancy engineering infrastructure such as water conveyance structures, water retaining structures, and water release structures. In addition, landslides can cause serious damage to life, property, and the environment, and have negative impacts on production, life, and ecology. The frequent

occurrence of landslides has had a profound impact on the normal operation of water conservancy projects and the ability to guarantee water quality [2].

Recent literature highlights the need for more efficient and accurate approaches to landslide disaster management, with remote sensing data and physical dynamics being identified as promising areas for research. However, there is still a need for methods that can overcome the limitations of traditional monitoring methods and provide more effective solutions for landslide disaster management. In locations prone to natural disasters, such as landslides, monitoring these events is a crucial tool for obtaining real-time and accurate data on the situation. Unmanned Aerial Vehicles (UAVs) have become one of the most common methods used for landslide monitoring. In disaster areas, UAV monitoring is commonly used to collect elevation DEM data and remote sensing image data, which serves as the national three-dimensional data foundation for achieving the integration of air and earth. Remote sensing photographs are also used to gather information about disasters, and this has become an inevitable trend in technological innovation. Monitoring landslides can overcome the limitations of topography, climate, and observation conditions. It can also perceive the elevation, locations of sliding mass, earth volume, landslide scope, and area before and after the landslide [3]. This information is critical for understanding the extent of the disaster and for developing effective prevention and treatment strategies. The reconstruction and visualization of the historical process of landslide catastrophes in complicated terrain environments based on landslide monitoring has become a high priority in the prevention and reduction of landslide disasters. Landslide monitoring reconstruction and visualization of the historical process of landslide hazards are important for understanding the causes of landslides, improving prediction and prevention, informing emergency responses, and developing better engineering solutions. Dynamic visualization of landslide disasters can enhance human perception and recognition of disasters, strengthen the information management capability of landslide emergency, and play a crucial supporting role in landslide disaster treatment decision making [4].

This study proposes a visual reconstruction method for the landslide process based on pre- and post-disaster remote sensing data. The objectives of this research are to develop a visual simulation of the landslide process and provide a scientific basis for landslide emergency response and subsequent treatment. The method is built upon the landslide physical dynamics and utilizes remote sensing data to reconstruct and visualize the landslide event. Specifically, we focus on the key technologies and methods of landslide spatial-temporal process dynamics reappearance and landslide disaster dynamic prediction based on pre-disaster and post-disaster remote sensing data. By integrating physical dynamics and remote sensing data, the proposed method can help overcome the limitations of traditional monitoring methods and provide a more effective solution for landslide disaster management. The goal is to achieve a visual simulation of the landslide process and provide a scientific basis for landslide emergency response and subsequent treatment.

The significance of this research lies in its potential to improve landslide disaster management and reduce the impact of landslides on human lives and infrastructure. By providing a more accurate and comprehensive picture of the landslide event, this method can inform decision making about disaster response, risk assessment, and mitigation efforts. Overall, this study contributes to the development of better, more efficient approaches to landslide disaster management, with the ultimate goal of reducing the impact of landslides on human lives and infrastructure.

## 2. Literature Review

The proposed framework integrates landslide dynamic modeling with the reconstruction and visualization of a landslide event. Consequently, we review related literature on two aspects: (1) landslide modeling; and (2) landslide visualization.

### 2.1. Landslide Modeling

Landslide modeling is an essential tool for predicting the occurrence and behavior of landslides, and has been the focus of a significant amount of research in recent years. Depending on the underlying modeling principles and mechanisms, landslide modeling can be partitioned into three categories.

Physically-based models [5] are widely used in landslide modeling due to their ability to take into account the physical processes that govern landslide behavior. These models incorporate equations describing the mechanics of slope failure, including the effects of gravity, soil properties, and groundwater flow. Such models can provide detailed information about the initiation and propagation of landslides, and can be used to simulate the effects of different environmental conditions and triggering events [6–8]. However, these models require extensive data on soil properties, topography, and other factors, which can be difficult and time consuming to obtain.

Empirical models [9–11] are simpler and more computationally efficient than physically-based models, and are often used when data are limited or when a quick assessment of landslide susceptibility is required. These models are based on statistical relationships between landslide occurrence and various environmental factors, such as slope angle, vegetation cover, and rainfall intensity. However, empirical models may not accurately capture the underlying physical processes that govern landslide behavior, and their predictive power may be limited in areas with different environmental conditions than those used to develop the model.

Hybrid models [12,13] attempt to combine the strengths of physically-based and empirical models, by incorporating both physical principles and statistical relationships into the modeling process. These models can provide more accurate predictions of landslide occurrence and behavior than purely empirical models, while requiring less data and computational resources than physically-based models. However, hybrid models may be more complex and difficult to implement than either purely empirical or physically-based models.

Despite advances in landslide modeling, there are still significant challenges to be addressed. These include the need for more accurate and comprehensive data on soil properties, topography, and other factors; the need to improve our understanding of the underlying physical processes that govern landslide behavior; and the need to develop models that can be easily implemented and provide timely and accurate predictions of landslide occurrence and behavior.

### 2.2. Landslide Visualization

The visualization technology of landslide dynamic evolution has been widely studied. The main research directions are as follows. The first is the automatic drawing method of landslide evolution map based on deep learning. Some scholars build a remote sensing system for landslide image processing evolution analysis by using various remote sensing data types related to ground deformation monitoring, optical data, landslide assessment, forest fire post-burn area assessment, critical infrastructure monitoring [14,15], space-borne interferometric synthetic aperture radar inversion technology [16,17], etc. The remote sensing system for image processing and evolution analysis of landslides can provide accurate and high-resolution terrain information of the landslide area, including measuring the surface deformation of the landslide area with millimeter-level precision and information about the temporal and spatial distribution of landslides. It can perform a detailed analysis of terrain changes and detect edge Poe's subtle movements and identify potential hazards. Furthermore, it can realize the continuous monitoring of large-area and long-term landslides, which is crucial for evaluating the evolution of landslides, understanding the mechanism and triggering factors of landslides, and predicting future movements. Based on this, some scholars [18] used the high-resolution optical satellite image based on cable for landslide detection, which is more reliable in identifying the hidden areas of a landslide hazard, and then carried out large-scale landslide detection with high precision and high

timeliness. An improved U-Net model is proposed to segment the landslide semantics of EO (Earth Observation) data at regional scale by using feature extraction blocks of CNN landslide mapping. However, there are some limitations and challenges in the landslide reconstruction system based on remote sensing, including that it may be affected by atmospheric conditions such as clouds, rain, and atmospheric turbulence, which will reduce the quality and accuracy of the data; multiple radar images need to be collected at different times. These images may be affected by changes in viewing geometry, radar system, or environmental conditions, making the accurate comparison and analysis of data challenging; the processing and analysis of remote sensing data can be computationally intensive and time consuming, requiring specialized software and technical expertise; remote sensing data may not provide sufficient information on the geological and hydrological conditions of landslide areas, which may limit the understanding of landslide mechanisms and triggers. Their operating costs can therefore be high, making them challenging to implement in developing countries or in resource-limited settings.

Another strand of methods combines the simple two-dimensional landslide model with the experimental test, and then constructs the three-dimensional prediction visual model. For example, researchers [19] obtained two-dimensional simulation results of a landslide area, the dip angle and influence depth based on a two-dimensional map of landslide monitoring. Other researchers [20] combined field monitoring with experimental tests and established a three-dimensional visualization model of landslide form prediction. However, the simple two-dimensional landslide evolution is relatively limited in time and space; therefore, the construction condition of a 3D prediction visualization model is relatively harsh. This includes information about soil properties, topography, weather conditions, etc. Developing a 3D visualization model is expensive, especially for large-scale landslides. The cost of collecting data, modeling, and analyzing results can be prohibitive for some organizations. Moreover, the method is difficult to test and the simulation prediction model is difficult to contain microscopic information.

The third category is the refined visualization method for the landslide simulation of a post-disaster state based on multi-source data fusion [21]. While techniques that focus on post-disaster results and analysis are important, pre-disaster information is also critical for understanding the factors that lead to landslide disasters. Earthquakes, geological conditions, heavy rain, and other environmental factors can all contribute to pre-landslide disasters, and are essential for disaster analysis and research. However, integrating data from multiple sources can be challenging, particularly when the data are in different formats or have varying levels of detail. The process of collecting, processing, and integrating data can be time consuming, which can delay the availability of simulation results. This can be problematic if stakeholders require timely information to make decisions. Therefore, it is necessary to develop a real-time and time-efficient landslide dynamic evolution method that integrates pre-disaster data with post-disaster analysis. This requires the development of sophisticated data processing techniques that can handle different data formats and levels of detail. The method should also be designed to provide timely information to stakeholders, allowing them to make informed decisions in a timely manner.

In summary, it is essential to develop an accurate, quick, and data-efficient dynamic evolution approach for landslides [22–28]. On the one hand, the multi-dimensional synthesis dynamically expresses the changes in spatial position, geometric form, and attribute information of geographical objects in the time dimension. On the other hand, it vividly and intuitively reveals the evolution of the geographical spatial-temporal process, to better reveal the relationship of landslide mechanics, acquire new knowledge, and discover the law. After the landslide occurred, the visualization simulation and analysis of the landslide process was carried out in the virtual geographic environment in the first time to realize data reorganization and rapid understanding of the disaster situation, so as to provide a scientific basis and important data support for disaster emergency treatment and further assessment and prediction.

## 3. Data Description

A massive landslide occurred at 21:20 on 23 July 2019, in the Chagou Formation of Pingdi Village, Jichang Town, Shuicheng County, Guizhou Province. The plane shape of the landslide is a long strip. After the landslide happened, the two original gullies along the slope were shoveled and scraped, which further accelerated the landslide process. As a result of the higher topography blocking some of them, the remainder transformed into a high-speed debris flow that impacted the residential areas on the hill, affecting a total of 23 families, 77 persons, and 27 dwellings, with 23 individuals buried. As of 29 July, there have been 43 fatalities and 9 missing persons.

Shuicheng County is located in the western part of Guizhou Province (N $26°10'$, E $104°35'$), with a terrain that generally slopes from northwest to southeast. Shuicheng County is situated in the northwest–southeast trending deformation zone of western Guizhou, also known as the Shuicheng–Ziyun deformation zone. This zone stretches for about 250 km in length and 20–50 km in width, with an overall trend of north $50°$ west to south $50°$ east. The landslide belongs to the tectonic erosion and denudation mid-mountain topography, with the highest elevation of the mountain behind the landslide being 2050 m and the elevation at the front valley being 1100 m, with a relative height difference of over 900 m. The regional topography is undulating, and the shape of the landslide is similar to a "long boot" shape, controlled by the northwest–southeast trending mountain on the northwestern side and the single-sided exposure to the east. The slope direction is $17°$ and the slope gradient is $30°$. The exposed rock types in the landslide area are, from youngest to oldest: Quaternary residual accumulation of gravel and clay, with a thickness of 3–6 m; Triassic Jialingjiang Formation shallow gray mudstone, Feixianguan Formation purple-red sandstone mudstone; Permian Xuanwei Formation (C11) gray-yellow sandstone and pink sandstone, Permian Emeishan basalt dark gray basaltic intrusive rock and volcanic intrusive rock, with a strong surface weathering and a relatively fragmented rock.

One month before the landslide occurred, the cumulative rainfall reached 371.9 mm, which was 1.6 times the average rainfall for that month in previous years. The maximum daily rainfall occurred on 12 July, with a value of 83 mm. In the week before the landslide (17–23 July), rainfall was relatively dense, with a cumulative rainfall of 153.7 mm and the maximum hourly rainfall occurring at 14:00 on 18 July, with a value of 26.3 mm/h. Statistical analysis of the rainfall data suggests that the previous period of heavy rainfall and short-term heavy rainfall may have been the main triggering factors for the landslide instability.

Figure 1 is the remote sensing image of the Chagou Formation before the landslide event. The elevation of the peak is 2083 m above sea level, and the bottom of the slope is 1210 m above sea level. The height difference reaches 873 m within the horizontal distance of 2690 m. The landslide erupted on a slope around 1540 m above sea level, 330 m from the bottom.

Figure 2 is the remote sensing image of the Chagou Formation after the landslide. According to the indications left by the landslide, this slope is not composed of stones, but rather of loose dirt. Prior to the landslide, it rained consistently on 21–23, with significant precipitation on 23 July. Consistent rainfall caused the soil water to melt, destabilizing the soil and triggering a debris flow, which resulted in a large and highly dangerous landslide.

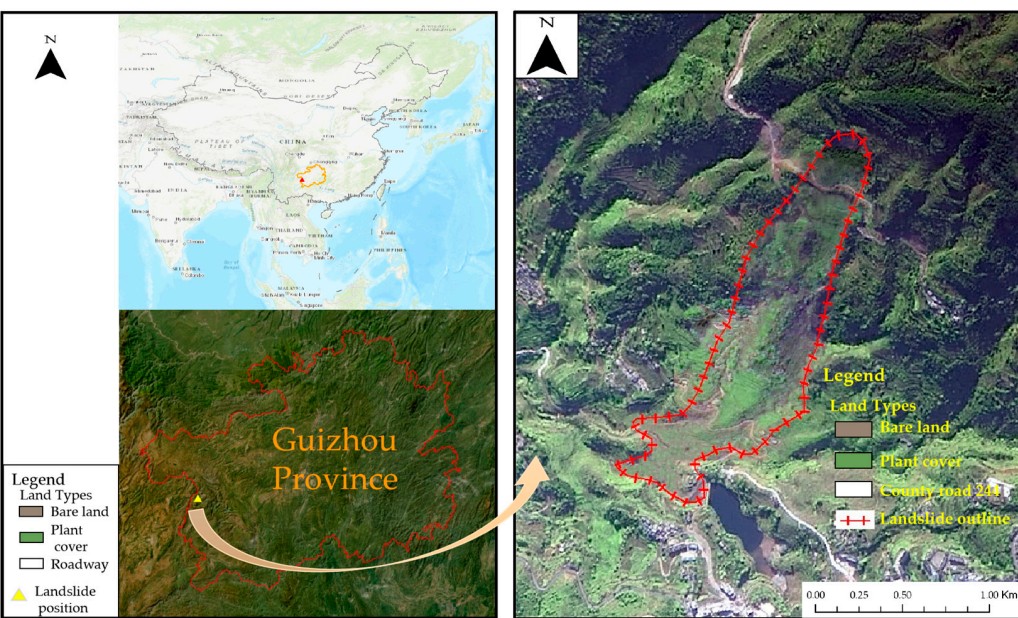

**Figure 1.** Location map and pre-disaster remote sensing image of landslide location.

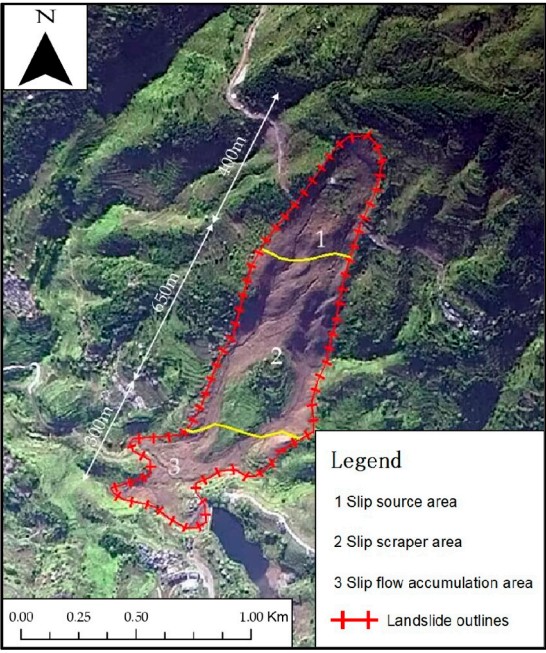

**Figure 2.** Post-disaster remote sensing image of landslide location.

## 4. Landslide Dynamics Modeling

The occurrence and evolution of a landslide depend on many factors, among which the dynamic factor and the mechanical property that determines whether a slope material will landslide or not are the most crucial determinants. The dynamic factor is controlled by the law of natural action, and the most important action is the driving force produced by gravity and atmospheric precipitation. The mechanical properties of the slope are composed of the mechanical properties of the slope material itself and the mechanical properties of the environment.

The two important factors controlling the occurrence and evolution of landslides are not constant natural scientific quantities, but two variables that are constantly changing. Gravity in the dynamic factors can be regarded as a constant measure, and its value is given by $F_0 = mg$, where $m$ is the mass of sliding mass and $g$ is the gravitational acceleration.

However, the dynamic force produced by atmospheric precipitation is a variable force: the precipitation impulse obtained by the mass $m$ of the sliding mass per unit volume in a unit time $t$ is

$$\Delta F_1 = q_1^2 t \text{ N} \tag{1}$$

where $q_1$ represents the amount of atmospheric precipitation passing through the space where the sliding mass is located per unit volume. The driving force of surface water flow formed by atmospheric precipitation, seepage water generated during rainfall, groundwater, debris flow, etc., is proportional to the water flow or debris flow passing through the landslide area, that is,

$$\Delta F_2 = \frac{\rho q_2^2 t}{S} \text{ N}, \tag{2}$$

where $\rho$ is the density of the fluid; $S$ is the cross-sectional area of the sliding mass receiving fluid impact per unit volume. Since the area per unit volume is equal to 1, $S = 1$. Therefore, the driving force controlling the landslide is a variable force:

$$F_t = F_0 + \Delta F_1 + \Delta F_2 = mg + q_1^2 t + \rho q_2^2 t \text{ N}. \tag{3}$$

Therefore, the driving force controlling the phenomenon of landslide motion can be characterized as

$$F_t = F_0 + \epsilon t \text{ N}. \tag{4}$$

In Equation (4), $F_0$ and $F_t$ are the initial driving force and terminal driving force of the sliding mass, respectively; $\epsilon$ represents the changing rate of the driving force; $t$ is the time.

The mechanical property of the slope is the second most important factor that specifies the occurrence of the landslide. Generally, the progress of natural action, the change (movement or deformation) in the slope material, and the characteristic value of the slippery property of the slope quality change with time. Therefore, the slidability eigenvalue also changes with time. Without loss of generality, let $E_0$ be the initial eigenvalue of the mechanical property, and let $\beta$ be the changing rate of mechanical property. The slidability eigenvalue $E$ over the time horizon is captured by

$$E_t = E_0 + \beta t. \tag{5}$$

Consider a slope with the slope angle $\alpha$. The landslide event can be simplified as a sliding mass moving along the slope. As mentioned before, the landslide event is determined by time-varying driving force $F_t$ and slidability eigenvalue $E_t$. Below we model the dynamics of the landslide and establish the dynamic equations of landslide events.

Without loss of generality, let $x_t$, $v_t$, and $a_t$ be the moving distance of the sliding mass, sliding velocity of the sliding mass, and the sliding acceleration of the sliding mass, respectively. As shown in Figure 3, the driving force $F$ and the resistance force $R$ determines the joint force $\hat{F}$, which is given by

$$\hat{F} = \sqrt{F^2 + R^2 - 2FR\cos \alpha} \text{ N}. \tag{6}$$

Note that the resistance force $R$ relates to the driving force through the slidability eigenvalue $E$ as

$$R = (1 - E)F \text{ N}. \tag{7}$$

Plugging Equation (7) into Equation (6) leads to

$$\hat{F} = F\sqrt{2(1 - \cos \alpha)(1 - E) + E^2} \text{ N}. \tag{8}$$

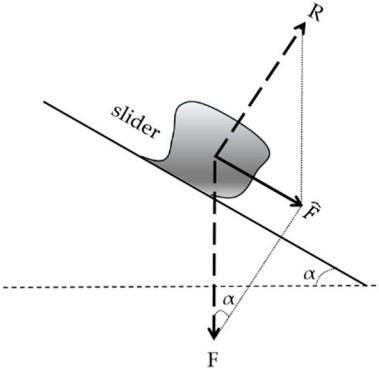

**Figure 3.** The schematic diagram of the sliding mass on the slope.

Equation (8) indicates that there is a unified relationship between the joint force, the driving force and the slidability eigenvalue. Note that both the driving force $F$ and the slidability eigenvalue $E$ are time varying. Substituting Equations (4) and (5) into Equation (8) leads to the time-varying joint force as follows:

$$\hat{F} = (F_0 + \epsilon t)\sqrt{2(1 - \cos\alpha)(1 - E_0 - \beta t) + (E_0 + \beta t)^2} \text{ N.} \qquad (9)$$

By Newton's second law of motion, the sliding acceleration is determined by

$$a_t = \frac{\hat{F}}{m} = \frac{(F_0 + \epsilon t)\sqrt{2(1 - \cos\alpha)(1 - E_0 - \beta t) + (E_0 + \beta t)^2}}{m} \text{ m/s}^2. \qquad (10)$$

Given the acceleration $a_t$, the sliding velocity $v_t$ can be characterized as

$$v_t = v_0 + \int_0^t a_t dt \text{ m/s.} \qquad (11)$$

Moreover, the sliding distance $x_t$ is given by:

$$x_t = \int_0^t \left[ v_0 + \int_0^t a_t dt \right] dt \text{ m.} \qquad (12)$$

There are four model parameters, namely $F_0$, $E_0$, $\epsilon$, and $\beta$, in the dynamic equations. To reconstruct and visualize the landslide event, we need to calibrate these model parameters. Below, we calibrate them based on the pre- and post-disaster remote sensing data.

For the initial driving force $F_0$, we can estimate the volume of the sliding mass from the actual landslide scenario and calculate the mass of the sliding by multiplying with the density of the sliding mass, i.e., $F_0 = mg$. For the remaining three model parameters $E_0$, $\epsilon$, and $\beta$, we calibrate them based on the pre- and post-disaster remote sensing data. Note that at the stationary state of the landslide, we have $v_t = 0$ and $a_t = 0$. Moreover, the sliding distance can be estimated by comparing the pre- and post-disaster remote sensing data, and the duration of the landslide is given by the difference between the timestamps of the pre- and post-disaster remote sensing data. Without loss of generality, let $\hat{x}$ be the sliding distance estimated from the pre- and post-disaster remote sensing data, and let $\hat{t}$ be the time duration of the landside event. $E_0$, $\epsilon$, and $\beta$ can be calibrated by solving the following equations:

$$\begin{cases} \int_0^{\hat{t}} \left[ v_0 + \int_0^t a_t dt \right] dt = \hat{x} \\ v_0 + \int_0^{\hat{t}} \frac{(F_0 + \epsilon t)\sqrt{2(1 - \cos\alpha)(1 - E_0 - \beta t) + (E_0 + \beta t)^2}}{m} dt = 0 \cdot \\ \frac{(F_0 + \epsilon\hat{t})\sqrt{2(1 - \cos\alpha)(1 - E_0 - \beta\hat{t}) + (E_0 + \beta\hat{t})^2}}{m} = 0 \end{cases} \qquad (13)$$

Equation (13) involves three unknowns and three equations. Therefore, we can obtain unique solutions for $E_0$, $\epsilon$, and $\beta$. Newton's method can be used to numerically solve the equation.

Finally, given the calibrated model parameters $F_0$, $E_0$, $\epsilon$, and $\beta$, we can simulate the movement of the sliding mass and reconstruct the landslide event based on the dynamic Equations (10)–(12).

## 5. Visualization

According to the comparison and analysis of the site survey and remote sensing images, the main sliding time is about 3 min, the main sliding direction is N 17° E, the sliding mass in the high-instability state is about $70 \times 10^4$ m$^3$, and the average overall downward dislocation is 50 m. The residual sliding mass in the slip source region is about $20 \times 10^4$ m$^3$, and the average overall downward dislocation is 0.4 m. The slip body that has left the slip source region is about $50 \times 10^4$ m$^3$, with an average downward dislocation of 2.2 m. Accordingly, dynamic characteristic parameters of sliding masses with different volumes are obtained according to Equation (13), as shown in Table 1. According to the dynamic parameters of the sliding mass with different volumes, the movement displacement of the sliding mass at different times is obtained from Equation (12), as shown in Table 2.

**Table 1.** Dynamic parameters of the sliding mass with different volumes.

| Sliding Volume | $E_0$ [1] | $\epsilon$ [2] | $\beta$ [3] |
|---|---|---|---|
| $20 \times 10^4$ m$^3$ | 0.5393 | 304.2114 | 1.8921 |
| $50 \times 10^4$ m$^3$ | 0.1152 | 6.4018 | 0.1711 |
| $70 \times 10^4$ m$^3$ | 0.0256 | 0.2326 | 0.0101 |

Note: [1] is the characteristic value of slippery property of slope material; [2] is the rate of change of applied force; [3] is the rate of change of waste of slope material.

**Table 2.** Movement displacement of sliding mass at different time.

| Displacement(mm) / Time (s) | Sliding Volume | | |
|---|---|---|---|
| | $20 \times 10^4$ m$^3$ | $50 \times 10^4$ m$^3$ | $70 \times 10^4$ m$^3$ |
| 0 | 0 | 0 | 0 |
| 30 | 726 | 62 | 13 |
| 60 | 3243 | 130 | 27 |
| 120 | 14,688 | 756 | 172 |
| 150 | 26,250 | 1036 | 289 |
| 180 | $50 \times 10^3$ | $2.2 \times 10^3$ | $0.4 \times 10^3$ |

To realistically and dynamically display the surface information of the landslide at different moments during the evolution of the landslide, we visualize the landslide evolution process based on the calibrated dynamic equations. Specifically, we discretize the whole duration of the landslide event and calculate the sliding distance $x_t$ at different timestamps $t$ based on Equation (12). Given $x_t$, we can calculate the elevation of the slope area at different timestamps $t$ and reconstruct the evolution of the landslide event. Based on the elevation data of the slope area at different timestamps, we can dynamically 3D visualize the landslide evolution process as follows. First, we perform three-dimensional segmentation according to the raster hope data with buried depth, that is, generate the space data of the sliding mass; then we perform texture mapping on the accumulation surface of the sliding mass at different timestamps and then perform real-time rendering on the spatial data of the sliding mass with texture at different timestamps and visual display to realize the three-dimensional dynamic visualization effect of the landslide process. The detailed procedure of 3D visualization of the landslide process is shown in Figure 4.

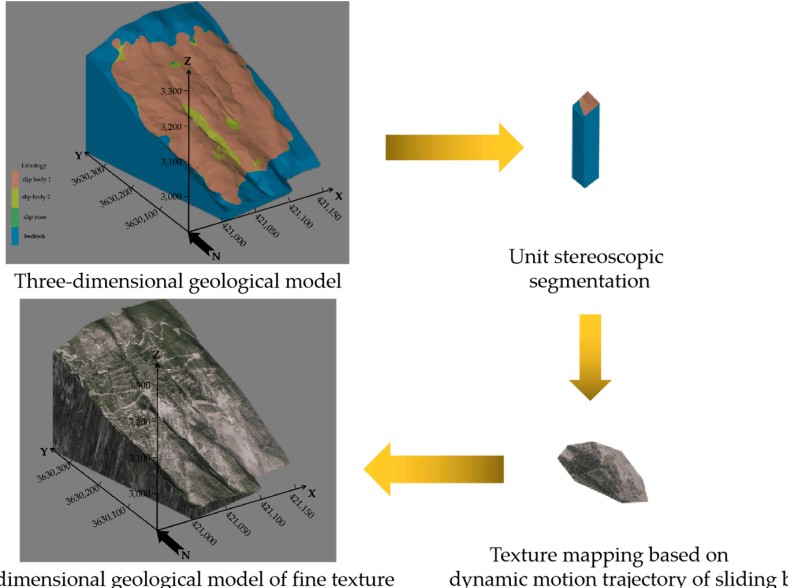

**Figure 4.** The procedure of 3D visualization of the landslide process.

Based on the realistic pre- and post-disaster remote sensing data of the landslide occurring at 20:40 a.m. on 23 July 2019 in the Chagou Formation of Pingdi Village, Jichang Town, Shuicheng County, Guizhou Province, we calibrate the model parameters in the dynamic equations, simulate and reconstruct the historical landslide evolution process, and dynamically visualize the landslide event following the procedure presented in Figure 4. The dynamic 3D visualization of the landslide event is demonstrated in Figure 5.

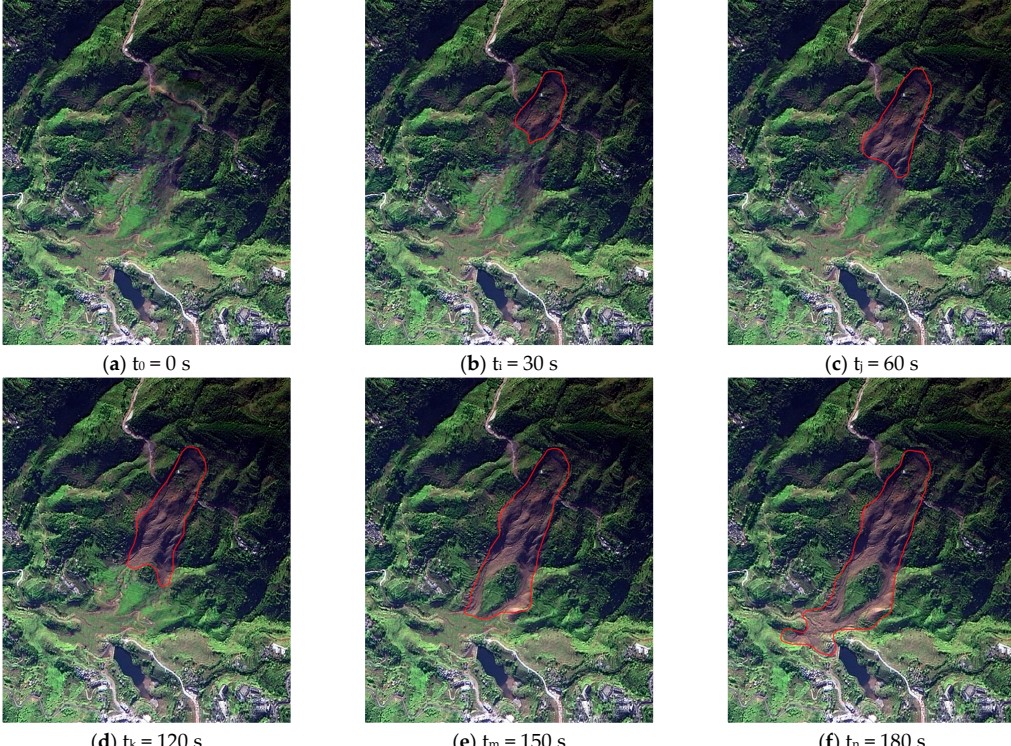

**Figure 5.** Dynamic 3D visualization of Chagou landslide event.

We can observe that the proposed approach realistically and dynamically reconstructs and visualizes the landslide event. Furthermore, the landslide process simulation can also detect the maximum burial depth, maximum sliding speed, maximum momentum, and

other indicators during the evolution process, and the visualization results can be used for subsequent risk assessment, engineering implementation, and other applications.

## 6. Conclusions

In this study, we proposed a landslide dynamic visualization method based on pre- and post-disaster remote sensing data, and applied it to the mountainous area of Guizhou, one of the most landslide-prone areas in China. The dynamic equations of the landslide event are established and calibrated based on realistic remote sensing data, and the spatial-temporal evolution process of the landslide event is reconstructed and visualized. Real-world case studies demonstrate that the proposed technique reconstructs and visualizes the landslide event accurately and dynamically.

The proposed approach of landslide reconstruction and visualization has extensive potential applications and societal benefits. First, it provides better understanding of landslide mechanisms. The physical dynamics method takes into account the mechanical behavior of the soil and rock, as well as the water content and slope gradient, which are all factors that contribute to landslide formation. By using this method, we can better understand the mechanisms that lead to landslides and make more accurate predictions of where they are likely to occur. Second, it enables a more accurate modeling of landslide processes. By combining the physical dynamics method with remote sensing data, we can create more accurate models of landslide processes. Remote sensing data, such as satellite imagery, can provide information on changes in terrain and vegetation cover that may indicate areas of instability. By incorporating these data into our approach, we can better predict the timing and magnitude of landslides. Third, it ensures improved emergency response. The pre-disaster and post-disaster visualization reconstruction method can help emergency responders to better understand the scope and scale of a landslide disaster. By visualizing the landslide process, they can identify areas that are most vulnerable and prioritize their response efforts accordingly. This can help to minimize the loss of life and property damage. Finally, it helps in better planning for future disasters. By using the physical dynamics method and remote sensing data to reconstruct the landslide process, we can identify areas that are most at risk for landslides in the future. This information can be used to develop better land use policies and engineering solutions that can help to prevent landslides from occurring in the first place.

The proposed method for reconstructing and visualizing landslide events based on pre- and post-disaster remote sensing data presents a promising approach for understanding the dynamic nature of landslides. However, there are several areas for future research and development that could further improve the accuracy and usefulness of this method. On the one hand, incorporating more factors and integrating other data sources could improve the method's accuracy. Our landslide dynamic modeling does not consider the detailed geological, geomorphological, geotechnical, or hydrogeological characteristics. Incorporating these features into the landslide modeling could improve its accuracy; however, it would also require a higher level of data. Therefore, incorporating data from ground-based sensors, such as seismometers or tiltmeters, could provide more detailed information about the behavior of a landslide and help to refine the model parameters. In addition, integrating real-time weather data into the model could help to predict the likelihood of future landslides and inform disaster response planning. On the other hand, the use of machine learning algorithms to improve the accuracy of the calibration process is explored. By leveraging large datasets of historical landslide events and associated remote sensing data, machine learning algorithms could be trained to automatically adjust the model parameters and improve the accuracy of the simulation results. Furthermore, there is an opportunity to explore the integration of this method with other hazard assessment and risk management tools. For example, the visualization results from the landslide process simulation could be integrated with flood models or building vulnerability assessments to provide a more comprehensive understanding of the potential impact of a landslide event.

This could inform land-use planning, disaster response planning, and infrastructure design and implementation.

In summary, while the proposed method for reconstructing and visualizing landslide events based on pre- and post-disaster remote sensing data presents a valuable approach, there are several areas for future research and development that could further enhance its accuracy and usefulness. With continued research and development, this method has the potential to improve our understanding of landslides and inform more effective hazard assessment and risk management strategies.

**Author Contributions:** Dynamics analysis and texture mapping principles, J.Y.; visual presentation, Y.G., W.C. and Q.M.; writing—original draft preparation, J.Y.; writing—review and editing, Z.L. and B.H. All authors have read and agreed to the published version of the manuscript.

**Funding:** This study was supported by the Open Research Foundation of Hubei Key Laboratory of Disaster Prevention and Mitigation (China Three Gorges University) (2021KJZ01).

**Institutional Review Board Statement:** Not applicable.

**Informed Consent Statement:** Not applicable.

**Data Availability Statement:** Not applicable.

**Acknowledgments:** The authors would like to thank all the reviewers for their detailed comments, which have greatly helped to improve the paper.

**Conflicts of Interest:** The authors declare no conflict of interest.

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
