# Peer review of "Reconstruction and Visualization of Landslide Events Based on Pre- and Post-Disaster Remote Sensing Data"

_water, doi:10.3390/w15112023_

Round 1
Reviewer 1 Report
I have reviewed the manuscript “Reconstruction and Visualization of Landslide Events Based on 2 Pre- and Post-Disaster Remote Sensing Data”. The proposed work has no characteristics that can define it as an essay. I think the manuscript needs a lot of improvement before it can be published. In some places it is unintelligible or difficult to read. I ask you to consider the following to improve the work and to re-submited as article.
INTRODUCTION
The chapter number is 1, not 0. The introduction lacks recent and adequate literature, problem statement/gap, the nobility, objective, and significance of the paper. It is unintelligible or difficult to read.
DATA DESCRIPTION
It is not a description of the data, but a poor description of the study area and the landslide. Completely rewrite with proper geographical, geological, geomorphological, tectonic, hydrological and climatic framework. Please supplement Figure 1 with the location of the study area in China. Please add the geological map of the area.
LANDSLIDE DYNAMICS MODELING
This model is treated in numerous works in the scientific literature. It is a very simplistic model where you do not consider any geological, geomorphological, geotechnical or hydrogeological parameters. Tackle the topic by discussing the limits and innovations and contextualizing it to your case study. Insert the appropriate citations on the updated literature.
VISUALIZATION
The approach used is interesting but needs to be explored and calibrated on the geological, structural and geomorphological characteristics of the landslide under study. What is the kinematics of the landslide? Did the landslide develop into a single event? How many slip surfaces are there? etc
DISCUSSIONS
Interpret your results and compare them to other similar works. I recommend integrating the bibliography on these aspects and making correlations with other case studies.
REFERENCES
The reference list has no consistency. Please integrate.
GENERAL COMMENTS
In my opinion, this paper is just an early draft of a research work that is worth publishing, after a thorough reading of impactful scientific literature, reframing the concepts, thinking, rewriting, editing.
I wish you good work!
Reviewer 2 Report
This manuscript describes an approach of integrating remote sensing observation with a modelling of landslides. The dynamic equations of the landslide event were calibrated based in remote sensing data.
The results are interesting, but it could be more fine if the application of these results on other site could be proposed, or give the essential information for its application.
Reviewer 3 Report
Basically, I would say this is a well-written paper. I only have some minor comments.
First, I suggest the literature review part could be divided into two parts, the first part is landslide modeling, and the second part is 3D visualization.
Second, insufficient references. There are many papers concerning 3D GIS or geovisualization in disaster management. Such as, DOI: 10.1080/13658816.2020.1833016, https://doi.org/10.1111/tgis.12922, https://doi.org/10.1016/j.aei.2010.05.008, https://doi.org/10.1016/B978-008044531-1/50437-1, https://doi.org/10.1680/cien.13.00020, please search for more papers and do a comprehensive review.
Third, the quality of most figures should be further improved.
Reviewer 4 Report
1. The landslide dynamics model should be further elaborated in particular no basic soil mechanics principles have been involved in the model.
2. It is hard to see the value of reconstruction/visualization of landslide events. As it is not a predictive tool, it is difficult to understand how the suggested approach can help land use or future planning as mentioned by the authors.
Reviewer 5 Report
This study aims at a demonstration system based on landslide visualisation. (i) investigating spatial geometrical morphological changes before and after landslides, (ii) coupling landslide motion states with degrees of freedom, and (iii) texture mapping.
The implementation of (i) to (iii) is the result of landslide visualisation. Once these techniques are established, simple monitoring that does not rely on instrumental measurements can be carried out, and this can be evaluated as a significant contribution to society. However, as this study is only a case study, it would be better to provide information with as much additional detail as possible, so that the information can be of greater benefit to society. Please add the following information.
1.Data Description(Cap2)
There is insufficient information on remote sensing datasets. What remote imagery was used in this study? This is important information to be returned to the reader.
2.Disscussion(Cap5)
The discussion in chapter 5 seems almost the same as in chapter 6, which is an abstract discussion. If the contribution is to be shown in disaster prevention monitoring in search of actual kinetic processes, I believe that chapter 5 should be presented with concrete examples of the research subject. There is a settlement directly under a landslide. It would be more effective to use this settlement as a case study to show to what extent risk can be avoided. If the results are not to include specifics, I feel that chapter 5 is unnecessary, as chapters 5 and 6 are almost identical in content.
Reviewer 6 Report
1.Every symbols used in Eqs. should be clearly stated, for instance, what are F1 and q2 in Eqs.1 and 2, respectively?
2. F(hat, the joint force) is missed in Fig. 3.
3.Why dose the resistance force R point upward?
4.Explain the relationships between F, F(hat), F1 and F2.
5.How do the authors handle the influence of vegetation on the DEM should be explained.
6.How to distinguish the influence of precipitation, surface runoff and the groundwater on the study landslide should be explained.
7.Some of my doubts may come from vague sentences, samantic problem, for instance, soil water to melt (Line 189), in the southeast of the northeast section of the Wumeng Mountains (Lines 170-171). Language editing is needed before resubmission.
8. It is better to provide unit to each symbol in equations.
9.The landslide area is huge. For Eq. 13, which point or movement mass is referred to determine the dynamic event should be indicated.
Round 2
Reviewer 1 Report
Analyzing the changes made, and having considered the responses to the reviewers I accept the work in its present form.
Congratulations to the authors for the work done
Reviewer 4 Report
Responses to my comments are noted and I have no further comments to add.
Reviewer 6 Report
The revised manuscript has addressed my comments appropriately.